Identification of the Golden-2-like transcription factors gene family in Gossypium hirsutum

Zhao Zilin 1 2
Shuang Jiaran 2
Li Zhaoguo 2
Xiao Huimin 2
Liu Yuling 2
Wang Tao 2
Wei Yangyang 2
Hu Shoulin 1
Wan Sumei 1 wansumei510@163.com
http://orcid.org/0000-0001-7916-2288 Peng Renhai 1 2 aydxprh@163.com
1 College of Plant Science, Tarim University , Alar, Xinjiang , China
2 Anyang Institute of Technology , Anyang, Henan , China
Ansari Mahmood-ur-Rahman
Electronic publication date: 2021 Nov 16
Publication date: 2021
Volume: 9
Electronic Location ID: e12484
Received 2021 Jun 8; Accepted 2021 Oct 22
Copyright: © 2021 Zhao et al.
Copyright year: 2021
Copyright holder: Zhao et al.
License: This is an open access article distributed under the terms of the Creative Commons Attribution License, which permits unrestricted use, distribution, reproduction and adaptation in any medium and for any purpose provided that it is properly attributed. For attribution, the original author(s), title, publication source (PeerJ) and either DOI or URL of the article must be cited.
License URL: https://creativecommons.org/licenses/by/4.0/

Keywords: Gossypium hirsutum, Golden2-like, Gene family, Phylogenetic analysis, Abiotic stress

Funding: National Natural Science Foundation of China 31471548 Central Plains Science and Technology Innovation Leader Project 214200510029 National Key Research and Development Program of China 2018YFD0100302 This work was supported by the National Natural Science Foundation of China (No. 31471548), the Central Plains Science and Technology Innovation Leader Project (No. 214200510029) to Renhai Peng and the National Key Research and Development Program of China (No. 2018YFD0100302) to Yuling Liu. There was no additional external funding received for this study. The funders had no role in study design, data collection and analysis, decision to publish, or preparation of the manuscript.

==============================
Background

Golden2-Like (GLK) transcription factors are a type of transcriptional regulator in plants. They play a pivotal role in the plant physiological activity process and abiotic stress response.

Methods

In this study, the potential function of GLK family genes in Gossypium hirsutum was studied based on genomic identification, phylogenetic analysis, chromosome mapping and cis-regulatory elements prediction. Gene expression of nine key genes were analyzed by qRT-PCR experiments.

Results

Herein, we identified a total of 146 GhGLK genes in Gossypium hirsutum, which were unevenly distributed on each of the chromosomes. There were significant differences in the number and location of genes between the At sub-genome and the Dt sub-genome. According to the phylogenetic analysis, they were divided into ten subgroups, each of which had very similar number and structure of exons and introns. Some cis-regulatory elements were identified through promoter analysis, including five types of elements related to abiotic stress response, five types of elements related to phytohormone and five types of elements involved in growth and development. Based on public transcriptome data analysis, we identified nine key GhGLKs involved in salt, cold, and drought stress. The qRT-PCR results showed that these genes had different expression patterns under these stress conditions, suggesting that GhGLK genes played an important role in abiotic stress response. This study laid a theoretical foundation for the screening and functional verification of genes related to stress resistance of GLK gene family in cotton.

Introduction

Transcription factors (TFs), also known as trans-acting factors, are known to activate or inhibit the transcription of downstream target genes at appropriate times. TFs affect all aspects of plant growth and development (Song et al., 2016; Ling et al., 2011). Plants are threatened by abiotic stresses during their growth, including high temperature, herbicides, heavy metals, drought, salinity, cold, pests and diseases (Nguyen et al., 2018). Crop yields are known to be seriously threatened under abiotic stress (Riechmann et al., 2000; Wray et al., 2003). Transcription factors can activate or inhibit gene transcription, affect the expression and function of their proteins, and play a significant role in plant stress response and physiological activities (Chrispeels et al., 2000; Mizoi, Shinozaki & Yamaguchi-Shinozaki, 2012).

For the first time, Golden2 (G2) has been identified that could cause maize to turn yellow (Jenkins, 1926). Subsequently, some studies have revealed that G2 protein was an important transcription factor in plant growth and development (Hall et al., 1998). Golden2-Like (GLK) transcription factors are members of the GARP superfamily (Riechmann et al., 2000; Xiao et al., 2019). Most GLK genes contain a Myb-DNA binding domain (which contains an HLH motif) and a C-terminal (GCT) box (Liu et al., 2017). In addition, some members of subgroups have conserved MYB-CC-LHEQLE domains (Qin et al., 2021).

GLK genes have been identified with different function in Physcomitrella patens, Arabidopsis thaliana, Oryza sativa, Capsicum annuum L. and Solanum lycopersicum (Hall et al., 1998; Rossini et al., 2001; Fitter et al., 2002; Waters et al., 2009; Yasumura, Moylan & Langdale, 2005). In terms of cell differentiation, functional redundancy of the GLKs is present in A. thaliana and rice (Chen et al., 2016). In corn, ZmGLK1 and ZmGLK2, a pair of homologous genes, have basically the same function and are expressed in mesophyll cells and vascular bundle sheath cells, respectively (Chang et al., 2012). These results indicate that GLKs are expressed differently in different photosynthetic cells, which control the process of cell differentiation and play an important role in chloroplast development (Yasumura, Moylan & Langdale, 2005). Studies have indicated that GLK1 is mainly expressed in leaf tissue and GLK2 is mainly expressed in fruit (Chen et al., 2016). In tomato, SlGLK2 is only expressed in the fruit, and changes the content of sugar and carotenoid by regulating chloroplast development, thus affecting the fruit quality of tomato (Powell et al., 2012; Nguyen et al., 2014). In addition, some studies have found that the KNOTTED1-LIKE HOMEOBOX (KNOX) gene acts on the downstream of SlGLK2 and only affects the chloroplast development in tomato fruit but not in leaf tissue (Nadakuduti et al., 2014). All these indicate that although GLK1 and GLK2 have the same function, they have different regulatory pathways in different organs and have tissue-specific characteristics (Nguyen et al., 2014). GLKs may enrich the carbon utilization rate of plants to a certain extent and promote the growth and exploitation of plants by increasing the fixation of carbon dioxide in root (Kobayashi et al., 2012). However, the expression of GLKs in leaves was significantly higher than that in roots (Fitter et al., 2002).

The GLKs also play an important role in biotic stress. At present, studies on GLKs in plant disease resistance mainly focus on A. thaliana, a model system, and some studies on rice, while few studies are known on other crops (Chen et al., 2016). The overexpression of AtGLK1 enhances the resistance of A. thaliana infected with Fusarium graminearum, and plays a positive role in the tolerance to Cucumber Mosaic Virus (Savitch et al., 2007; Schreiber et al., 2011; Han et al., 2016). AtGLK1 regulates genes related to disease resistance, and its effect on different pathogens are different (Chen et al., 2016). In addition to A. thaliana, OsGLK1 in rice has also shown certain disease resistance (Nakamura et al., 2009). In addition, GLKs are also involved in hormonal response. The resistance of AtGLK1 to Hyalo-peronospora arabidopsidis Noco2 involves two signaling pathways: salicylic acid and jasmonic acid (Murmu et al., 2014). The GLK1/2-WRKY40 transcription module plays a negative regulatory role in the ABA response (Ahmad et al., 2019).

Many functions of GLK genes have been studied in depth, but the functions related to abiotic stress are rarely mentioned and few studies have been reported. Research has found that some genes in the GLK gene family are involved in stress response in maize (Liu et al., 2016). It was found that the down-regulation of the SlGLK29 gene in tomato could reduce the cold resistance (Liu, 2018). This suggests that different GLK genes have distinct expression patterns under different stress condition.

Cotton is an important fiber and oil crop in the world, but its development is always threatened by abiotic stresses, including extreme temperature, drought and salt (Zhang et al., 2018; Li et al., 2021). These stresses will become more severe in the future, which will lead to lower crop yields and quality (Onaga & Wydra, 2016). Therefore, it is necessary to identify genes related to abiotic stress resistance to assist cotton genetic improvement. So far, the identification of the GLK gene family in G. hirsutum has not been reported. In our research, the GLK gene family in G. hirsutum was identified, and the subcellular localization, chromosomal distribution, gene structure and expression level of GLK genes were analyzed. The results demonstrate that nine key genes responded to drought, salt and cold stress. This study provides a reference for further study of the role of GLKs in the stress response of cotton.

Materials & methods

Identification of the GLK genes in Gossypium hirsutum

The genome files of G. hirsutum (TM-1 HAU_v1.1) and Theobroma cacao L. (assembly Criollo_cocoa_genome_V2) were obtained from CottonFGD (http://www.cottonfgd.org/) (Wang et al., 2019) and NCBI (https://www.ncbi.nlm.nih.gov/) (Argout et al., 2011). The genome and protein sequences of A. thaliana were obtained from NCBI (https://www.ncbi.nlm.nih.gov/). The Markov Model (HMM) of PF00249.31 and PF14379.6 were downloaded from Pfam (https://pfam.xfam.org/). The TBtools v1.087 HMM Search (Chen et al., 2020) was used to identify the GLK gene family in A. thaliana, G. hirsutum and T. cacao with PF00249.31 and PF14379.6, which were most likely members of GLK gene family. Published tomato GLK protein sequences were downloaded from NCBI (https://www.ncbi.nlm.nih.gov/) (Liu, 2018). The A. thaliana and tomato sequences were used as query sequences to BLAST against the cotton and T. cacao genome database using TBtools (Chen et al., 2020). ProtParam tool (https://web.expasy.org/protparam/) was used to predict the physical and chemical properties of GLK proteins, including the number of amino acids, molecular weight (MW), isoelectric point (PI), and Instability index. The subcellular localization of GLKs was predicted through the WoLF PSORT (https://wolfpsort.hgc.jp/) and CELLO v.2.5 (http://cello.life.nctu.edu.tw/) resources (Yu et al., 2006).

Phylogenetic analysis of GLK genes

A phylogenetic tree was constructed the obtained GLK protein sequences of A. thaliana, tomato, T. cacao and G. hirsutum. A neighbor-joining tree of GLK genes was constructed using MEGA-X with 1,000 bootstrap replications (Hall, 2013). The phylogenetic tree was drawn using EvolView (He et al., 2016).

Analysis of the conserved motifs and gene structure

The conserved sequences of GLK was identified and analyzed by the MEME website (http://meme-suite.org/) (Bailey et al., 2009). The optimal width was 10 to 150, and the number of motifs was 26. Everything else was set to default values. TBtools (Chen et al., 2020) was used to map conserved motifs and gene structures (introns and exons), using MAST profiles from the MEME website and the GFF3 profiles for each gene.

Chromosomal localization analysis of GLK genes

Based on the genome and genome annotation files of G. hirsutum, the chromosome distribution of GLK genes and their physical locations were obtained through TBtools (Chen et al., 2020).

Cis-regulatory elements analysis of GLK

In order to analyze the promoter of GLK in G. hirsutum and predict the function of the GLK genes, 2,000 bp sequence upstream of the start codon for each gene was extracted and input into the Plantcare website for analysis (http://bioinformatics.psb.ugent.be/webtools/plantcare/html/).

GO and KEGG enrichment analysis of GLK genes

For functional enrichment analysis, GhGLKs were submitted to the omicshare tool (https://www.omicshare.com/tools/) for GO and KEGG enrichment analysis.

Differential gene expression analysis

In order to study the expression pattern of the GLK in G. hirsutum, the transcriptome sequencing data of G. hirsutum (PRJNA490626) under cold, salt, and drought stress was downloaded from NCBI (https://www.ncbi.nlm.nih.gov/). Trimmomatic (Bolger, Lohse & Usadel, 2014) was used to remove the adapter and perform quality control. Reads were mapped to the genome using the hisat2 program (Kim, Langmead & Salzberg, 2015), and then Fragments Per Kilobase of transcript per Million fragments (FPKM) values of GLK genes were calculated by Cufflinks (Ghosh & Chan, 2016; Pollier, Rombauts & Goossens, 2013). The expression level of GLK family genes was standardized, expressed as a FPKM value, and transformed in log2 form, and the heatmap was drawn using TBtools software (Chen et al., 2020).

Stress treatments and qRT-PCR analysis

G. hirsutum acc. TM-1 planted in the experimental field of Anyang Institute of Technology was selected as experimental material. The seedlings with stable growth of G. hirsutum were treated with three stresses: salt stress, drought stress and cold stress. The root of cotton seedlings was irrigated with 250 mM NaCl solution and 18% PEG solution to simulate salt stress and drought stress. It was placed in a 4 °C incubator to simulate cold stress treatment. Leaves from cotton seedlings with consistent growth were collected after 0, 1, 3, 6, 12, 24 h of above stresses. All samples were immediately frozen in liquid nitrogen and stored at −80 °C for RNA extraction.

Total RNA was extracted from each sample using the EASYspin Plus Plant RNA Kit (RN38, Aidlab Biotech, Beijing, China). The quality of RNA was determined by agarose gel electrophoresis and a Nanodrop2000 nucleic acid analyzer. The cDNAs were synthesized using a TranScript All-in-One First-Strand cDNA Synthesis SuperMix for qPCR (Transgen Biotech, Beijing, China). The kit used for Real-time PCR was the TransStart Top Green qPCR SuperMix kit (Transgen Biotech, Beijing, China). The instrument used was the ABI 7500 Fast Real-Time PCR system (Applied Biosystems, Waltham, MA, USA). The specific primers for these differentially expressed genes were designed using the Prime-Blast in the NCBI online database and were listed in Table S1. Each experiment was repeated three times, and chose two groups of good data to graph. The relative gene expression levels were analyzed using the 2−ΔΔCt method (Livak & Schmittgen, 2001).

Results

Identification and analysis of basic physicochemical properties of GLK family members in G. hirsutum

A total of 146 GLK genes were identified from G. hirsutum, named GhGLK1-GhGLK146. The basic physical and chemical properties were also predicted and analyzed. Amino acid sequences range in length was from 140 (GhGLK27) to 826 (GhGLK139), the isoelectric point was from 5.07 (GhGLK9) to 9.71 (GhGLK134). The instability index refers to how stable the protein was in the test tube (≤40, possibly stable; >40, possibly unstable). Prediction showed that, except for 13 genes, all other genes may be stable. According to the results of subcellular localization, all the GhGLK genes were located in the nucleus. GhGLK6, GhGLK46, GhGLK76, GhGLK128 and GhGLK139 were located in the chloroplast and nucleus. GhGLK112 and GhGLK127 were in cytoplasm and nucleus (Table S2).

Phylogenetic analysis of GLK in G. hirsutum

Using tomato, A. thaliana and T. cacao GLK protein sequences as reference, a rootless phylogenetic tree of GLK protein in G. hirsutum was constructed (Fig. 1). As the results shown, 146 GLK genes of G. hirsutum were divided in 10 subfamilies. At the same time, the number of GLKs in G. hirsutum was much higher than that in A. thaliana, T. cacao and tomato.

Figure 1 Phylogenetic analysis of GLK proteins from Gossypium hirsutum, Theobroma cacao L., tomato and Arabidopsis thaliana.

The blue marks are members of the tomato GLK family; The green circles are members of the A. thaliana GLK family; The purple squares are members of the T. cacao GLK family; The yellow star is the GLK member of cotton subgroup A, and the red triangle is the D subgroup. The blue squares are GLK genes located on the scaffolds.

Gene structure and protein conserved motif analysis of GLK in G. hirsutum

The motif of each protein, namely the conserved element, was analyzed by MEME. A total of 26 possible motifs were identified. Motif 1 was included in all GhGLKs, and it was conservative for GhGLKs. These motifs differ between subfamilies, but were conserved within each subfamily. For example, motifs 17 and 18 were found only in subfamily D and E, respectively. Some subfamilies had very conservative motifs. For example, all members of the C subfamily had motif 5, 3, 8, 2, 1, 14, 10. Predictions about exons and introns of the GLK genes were shown in the Fig. 2. The number of exons was at least one and at most eleven. Except for GhGLK87, the other members of the D subfamily had one exon, and all members of the C subfamily had 11 exons. Most of the other members had between 4 and 7 introns. The number of exons and introns varied among different subfamilies, but in the same subfamily, the exon-intron structure of most members showed great similarity.

Figure 2 The conserved motif and exon-intron structure of GLK genes in Gossypium hirsutum.

(A) Phylogenetic analysis of GhGLK genes. (B) Analysis of conserved motif of GhGLK protein sequences. Different motifs are shown in a specific color. (C) Intron and exon analysis of GhGLK genes. Exons and introns are represented by yellow boxes and thin lines, respectively. The UTR is shown in a green box.

Chromosomal localization analysis of GLK family genes in G. hirsutum

Based on a physical map of the GLK family members of G. hirsutum, 144 of 146 GLK genes were located on 26 chromosomes (Fig. 3). The other two genes were located on the scaffold. Among the chromosome-located genes, 17 genes were located on chromosome 5. There were 16 genes located on chromosomes 8 and 12, only six genes were located on chromosome 2 (Fig. 4). There were significant differences in At sub-genome and Dt sub-genome about the number and location of genes. For example, there were two genes on chromosome A02 and four genes on chromosome D02. This may be due to the fact that G. hirsutum is a tetraploid cotton species, which is a hybrid between an A-genome-like Gossypium herbaceum and a D-genome-like Gossypium raimondii.

Figure 3 Chromosome distribution statistics of GhGLK genes.

Blue represents the number of genes locate on each chromosome of subgroup A, while red represents the number of genes locate on each chromosome of subgroup D.

Figure 4 Location of GhGLK genes on chromosome.

GhGLKs are located on 26 chromosomes of Gossypium hirsutum, and two genes are located on scaffold. Chromosome names are shown on the left and gene names are shown on the right.

Analysis of promoters of GLK genes

The Plantcare was used to analyze the sequence of 2,000 bp upstream of the promoter region of GhGLK genes, and the found cis-regulatory elements were shown in Fig. 5. We found several cis-regulatory elements in stress response, which were associated with anaerobic induction, defense and stress response, drought, cold, and wound, respectively. Meanwhile, we also found a number of hormone-responsive cis-regulatory elements, which were associated with abscisic acid (ABA), auxin (IAA), gibberellin (GA), salicylic acid (SA), and methyl jasmonate (MeJA). Among all cis-regulatory elements, the number of elements related to light response was the largest and the distribution was the widest. In addition, we had identified cis-regulatory elements involved in cell cycle regulation, circadian rhythm, down regulation of photosin expression, and regulation of flavonoid biosynthesis genes. In summary, different types and quantities of cis-regulatory elements were distributed in different GhGLK promoters of cotton. According to the results, it was speculated that under environmental stress, the cis-regulatory elements leaded to the expression of GhGLK genes, thereby enhancing the resistance to environmental stress.

Figure 5 Analysis of cis-regulatory elements in promoters of GhGLK genes (A–J).

Cis-regulatory elements with the same function are shown in the same color.

GO and KEGG enrichment analysis of GLK in G. hirsutum

In order to further understand the function of GhGLKs, we carried out functional enrichment annotation of gene ontology (GO) and kyoto encyclopedia of genes and genomes (KEGG). The results improved our accurate understanding of gene function, including many significantly enriched terms (Fig. 6, Table S2). The GO enrichment analysis of GhGLKs was divided into three categories, including biological processes, molecular functions and cellular components, of which the biological process had the largest number of 13 GO entries, followed by molecular functions and cellular components. The biological process was mainly concentrated in four subclasses: cellular process, metabolic process, regulation of biological process and biological regulation. The molecular function mainly included binding, nucleic acid binding transcription factor activity and catalytic activity. The cellular components were mainly organelle subclasses. KEGG pathway enrichment analysis of GLK family genes revealed two pathways, RNA transport and plant hormone signal transduction respectively. To sum up, the functional enrichment analysis results confirmed the functions of GhGLKs in many biological processes, which were related to plant growth and development, plant hormone signal transduction and so on.

Figure 6 GO and KEGG enrichment analysis of GhGLKs.

(A) The numbers of level2 GO terms. (B) KEGG pathway enrichment analysis of GhGLKs.

Expression analysis of GLK gene in G. hirsutum under different abiotic stresses

To investigate the response of GhGLK to adversity stress, we analyzed the expression levels of 146 GhGLKs under different adversity conditions. The results showed that the expression of GhGLK genes altered under salt, drought and cold stress, which revealed that GhGLKs were involved in the regulation of adversity stress (Fig. 7). According to the above results, we selected nine key genes with differentially expressed GhGLKs and analyzed the expression pattern by qRT-PCR (Fig. 8).

Figure 7 Differentially expressed genes of GhGLKs under cold, NaCl and PEG stress.

Figure 8 Expression patterns of GhGLK gene family members under cold, drought, and salt stress.

(A) Expression patterns analysis of GhGLKs under salt stress. (B) Expression patterns analysis of GhGLKs under drought stress. (C) Expression patterns analysis of GhGLKs under cold stress.

Under salt stress, the expression levels of all GhGLK genes increased after 1 h, reached a minimum at 12 h, and increased again at 24 h. Among them, the expression levels of GhGLK38, GhGLK55, GhGLK82, and GhGLK120 increased significantly at a certain moment, reaching the highest level at 1, 24, 6, and 1 h respectively. Except for GhGLK1, GhGLK39 and GhGLK82, other genes increased after 1 h of treatment, then continued to decline, and increased again after 24 h. These genes had a certain trend under salt stress treatment, and the response to salt stress was more obvious, and the expression of genes also had certain differences.

Under drought stress, GhGLK1, GhGLK8, and GhGLK120 genes were significantly down-regulated after 1 h of treatment, and their expression levels were extremely low, indicating that these genes may play a negative regulatory role under drought stress. GhGLK46 and GhGLK55 genes changed significantly, both were up-regulated after 1 h of treatment and reached a peak at 3 h, decreased after 6 h, and then continued to increase. There was no significant change in the expression levels of the two genes GhGLK39 and GhGLK47.

Under cold stress, all GhGLK genes were significantly lower than that under drought stress and salt stress, which indicated that GhGLK genes were not strongly induced by low temperature stress. Except for GhGLK8, GhGLK38, GhGLK82 and GhGLK120, the other genes were up-regulated after 1 h of low temperature treatment. GhGLK55 was strongly induced by cold stress, and its expression continued to increase, reaching a peak at 3 h and beginning to decrease. GhGLK38 reached the lowest after 12 h of cold stress treatment, GhGLK39 and GhGLK82 reached the lowest at 3 and 1 h, respectively, and all other GhGLKs reached the lowest at 24 h.

Discussion

As the main cultivated cotton species in the world, G. hirsutum is extremely important to human life. Adversity coercions, including drought, salt and cold, are a very serious problem to plant growth and development. Golden2-Like is a type of transcription factor that exists widely in plants, and belongs to an important type of transcription factor in the GARP transcription factor superfamily of Myb transcription factors. It had been identified in many species and found to be closely related to abiotic stress, but it had not been reported in cotton.

Herein, a total of 146 GLKs were identified from G. hirsutum, named each gene according to its position on the chromosome. The number of GLK gene in cotton was much higher than that in tomato (54 genes) (Liu, 2018) and maize (59 genes) (Liu et al., 2016), which might be related to the relatively large genome of G. hirsutum heterotetraploid and the complexity of gene regulationdue to the complexity of the genome. The analysis of the physical and chemical properties of upland cotton GhGLK showed that its sequence length, relative molecular weight and isoelectric point distribution range were very large. This might be due to the large-scale replication of the upland cotton genome and the large number of upland cotton GhGLK genes. This study predicted the location of G. hirsutum GLKs in the cell, and found that some genes were located in the chloroplast. Previous studies had shown that the GLK gene is related to the growth and development of the chloroplast. This result was also consistent with the results of previous studies.

We analyzed the phylogenetic relationship of G. hirsutum GLKs, and constructed a phylogenetic tree of tomato, A. thaliana and G. hirsutum. The results showed that the GLK gene of G. hirsutum can be divided into 11 subfamilies, while tomato and corn were divided into six subfamilies (Liu, 2018). This might be due to the fact that the number of GLK genes in cotton was larger than that of GLK in tomato, resulting in a more precise and accurate sub-family classification. Some GhGLK and tomato GLK were homologous, and the results of this study indicated that there was a close evolutionary relationship between GLK genes. At the same time, we found that GhGLK gathered in the same subfamily had similar motif distribution patterns, motif positions and lengths.

The gene structure analysis also showed that the GLK gene had strong evolutionary conservation. GLK genes in the same subgroup had similar number of exons/introns and length of exons. The results of previous studies showed that the structure of the exons and introns of each subfamily of GLK genes in tomato showed a large similarity, and the number of introns of most members was between 4 and 7. The results were the same in G. hirsutum, except that there are up to 11 exons in upland cotton and up to seven in tomato (Liu, 2018).

In the G. hirsutum GLK gene family, 144 genes were located on 26 chromosomes of group A and D, and the other two genes were located on scaffolds of unknown chromosomes. Among them, there were 70 GLK genes on chromosomes of group A and chromosomes of group D. There were 74 GLK genes on it. The distribution of the GLK gene in G. hirsutum was uneven on the chromosomes, but the distribution of the GLK gene on the two homologous chromosomes was indeed the same.

Many cis-elements in GhGLK promoter were related to biotic and abiotic stress. In order to further understand the function of GhGLK under different environmental stresses, we analyzed the expression patterns of nine GhGLK under different environmental stresses by qRT-PCR. This showed that most selected GhGLKs respond to abiotic stresses, including drought, cold, and salt. In previous studies, it was found that the expression pattern of ZmGLK3 gene in maize was up-regulated under drought and salt stress (Liu et al., 2016); the expression of SlGLK7 in tomato was up-regulated under three stresses (Liu, 2018). The expression of SlGLK15 and SlGLK37 genes in tomato decreased first and then increased during cold stress and salt stress, and these two genes belonged to the same subfamily (Liu, 2018). In our results, we found that the two genes GhGLK55 and GhGLK120 were located in the same subgroup, and they were in the same subgroup as SlGLK15 and SlGLK37 in tomato. These two genes in G. hirsutum also decreased firstly and then increased under cold and salt stress, indicating that the orthologous genes they should have similar expression patterns in the face of adversity and coercion. We also found that the differentially expressed genes all have anaerobic-induced and light-responsive cis-acting elements, and they all contained at least one abiotic stress-responsive element. GhGLK120 had the most hormone-responsive elements.

Conclusions

In this study, 146 GhGLK genes were identified in G. hirsutum. Based on the analysis of their physical and chemical properties, subcellular localization, phylogenetic relationship, distribution of cis-regulatory elements, we identified nine key GhGLKs involved in salt, cold, and drought stress. The qRT-PCR results of these genes in three stress responses showed that GhGLKs played an important role in abiotic stress responses in cotton. It was of great significance to make full use of its resistance germplasm resources and provided a theoretical basis for further mining of cotton resistance genes.

Supplemental Information

Supplemental Information 1 Use NCBI designed differentially expressed gene-specific primer sequences.

Click here for additional data file.

Supplemental Information 2 Physico-chemical and biochemical characteristics of GLK genes in Gossypium hirsutum.

Click here for additional data file.

Supplemental Information 3 The data of Fig. 8.

qRT-PCR results of 9 GhGLK genes under salt, drought and low temperature treatments.

Click here for additional data file.

Additional Information and Declarations

Competing Interests

Author Contributions

Data Availability

The authors declare that they have no competing interests.

Zilin Zhao conceived and designed the experiments, performed the experiments, analyzed the data, prepared figures and/or tables, authored or reviewed drafts of the paper, and approved the final draft.

Jiaran Shuang performed the experiments, analyzed the data, prepared figures and/or tables, and approved the final draft.

Zhaoguo Li performed the experiments, analyzed the data, prepared figures and/or tables, and approved the final draft.

Huimin Xiao performed the experiments, analyzed the data, prepared figures and/or tables, and approved the final draft.

Yuling Liu performed the experiments, analyzed the data, authored or reviewed drafts of the paper, and approved the final draft.

Tao Wang analyzed the data, authored or reviewed drafts of the paper, and approved the final draft.

Yangyang Wei analyzed the data, authored or reviewed drafts of the paper, and approved the final draft.

Shoulin Hu analyzed the data, prepared figures and/or tables, and approved the final draft.

Sumei Wan conceived and designed the experiments, authored or reviewed drafts of the paper, and approved the final draft.

Renhai Peng conceived and designed the experiments, authored or reviewed drafts of the paper, and approved the final draft.

The following information was supplied regarding data availability:

The raw data are available in the Supplemental File.

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
