# Peer review of "Identification of the Golden-2-like transcription factors gene family in Gossypium hirsutum"

_PeerJ, doi:10.7717/peerj.12484_

## Round 0.1 · original submission · Major Revisions

Dear Corresponding Author,

Both the reviewers have raised serious issues in the paper under review, especially Reviewer-2. The said Reviewer has sent an additional file of comments to address which are very important while revising your manuscript. Please make sure:

1. A point-wise rebuttal is required for the reviewer's comments

2. Please also address the additional comments raised by Reviewer-2 which are attached in PDF format.

Reviewer 1 ·

Basic reporting

Zhao et al described the identification of GLK genes in Gossypium hirsutum using structural genomics approaches and discussed their potential functions in drought stress based on tissue-specific expression patterns, drought response (qPCR), and co-expression networks. The reviewer found that the methods and results are solid and may require only minor revisions.

Experimental design

The authors analyzed the GLK gene family in cotton using bioinformatics approaches. These analyses can help to understand gene function. This is good work, and well written.

Validity of the findings

For my part, this paper is prepared well. The authors used well-established methods to perform analysis. Results and data interpretation are comprehensive and reasonable. The conclusion is also a suggestive conclusion. However, there also some concerns that should be addressed before publication.

Additional comments

A professional English editor is needed to improve the language. Grammar mistakes and misspellings are mostly in the manuscript, which makes the manuscript difficult to read. The authors should check the manuscript very carefully before the submission; well I understand that English is not their native language.

·

Basic reporting

The general outline of the manuscript obeys the journal conditions, so to that far, the line is OK, however, the flow of thought is not proper, and it is paramount for the authors to polish the manuscript before resubmitting for a second evaluation.

Experimental design

Materials and methods
Databases: to be corrected to the database, only a single database is shown in the manuscript. Moreover, this section should be merged with “the genome-wide identification of the GLK genes in Gossypium hirsutum” .
The section for genomewide identification…. Should be corrected to state “Identification of the GLK genes in Gossypium hirsutum”
A serious concern of the use of the two protein domain “PF00249.31 and PF14379.6” in which PF00249.31 Myb_DNA-binding (PF00249) - Pfam: Family and PF14379.6: Myb_CC_LHEQLE; MYB-CC type transfactor, LHEQLE motif, and yet the gene or the TF family under study is the GLK: can the authors explain the mismatch, because this raises serious question of the identification of the gene or the TF under study.
Phylogenetic tree analysis: The authors have used protein sequences obtained from A. thaliana, tomato and G. hirsutum. The very first question which rings into the mind of the reader, what was the justification of using the protein sequences from these three plants to carry out the phylogenetic tree analysis, moreover, is tomato evolutionary closer to cotton compared to Theobroma cacao. The second serious concern is no information is provided within the manuscripts on how the protein sequences from Tomato and Arabidopsis were obtained.
This section is poorly done and must be redone, with protein sequences obtained from amore closer relatives to cotton.
Promoter cis-element analysis of GLK: Change to Cis-regulatory elements. Is true as pointed by the authors that through the determination of the cis-regulatory elements one is able to determine the function of the gene or the TF, the answer is no, so the authors need to rephrase.
Differential gene expression analysis: The authors need to simply state “RNA seq expression analysis” reason for doing this not stated not indicated, and furthermore, it is secondary data already in the public domain, unless the data was generated by the authors, this need to be clearly shown in order to improve the validity and novelty of the submitted manuscript
Stress treatments and qRT-PCR analysis: The treatment initiated “cold, salt and drought” and yet the RNA seq used as the reference point were profiled after the plants were exposed to “low temperature, salt and drought”, the question to the authors can low temperature be equated to cold stress?

Validity of the findings

Not valid, because in the materials and methods, serious questions have been raised, the results will only be valid if all the issues raised corrected.

Additional comments

Substantial revision is needed before the paper could be reviewed again, and if the authors are not able to carry out all the issues stated, I fully recommend rejection. Moreover, the manuscript is too shallow, if possible the authors could be advised to validate the key genes or the TFs as per the RNA seq. and RT-qPCR analysis.

---

## Round 0.2 · Minor Revisions

Instead of returning the manuscript to the reviewers, I have gone through the paper and am largely satisfied with the improvements the authors have made.

It is great to see that the paper under consideration is much improved. However, it need more effort to be acceptable. The authors are needed to consider the following comments:

1. The abstract should be in structured form. Please indicate different sections like Background, Methods, Results, etc.
2. Methods are missing in abstract section.
3. "many cis-regulatory elements" statement in abstract is misleading and confusion. Be clear how many cis-regulatory elements in different stresses were identified?
4. Last 3 lines of "Introduction" should be the part of "Discussion" or "Conclusion". Rather, here the hypothesis of study need to be mentioned.
5. Figure 1, 2, 4, 5 are of very low resolution and not readable. Please provide high resolution images.
6. In Figure 3, the legend on Y-axis may be "Number of genes" instead of "Gene number".
7. How many genes were subjected to Real Time PCR analysis? Apparently, there were 9 genes studied. What was the criteria to select 9 genes out of 146 genes?
8. There is no need to provide Table-1, as the same information is given as Supplementary Table 2. So delete Table 1.

---

## Round 0.3 · Minor Revisions

One of the section editor has raised the following concerns regarding manuscript. Please address these concerns before final decision is made:

"The manuscript was in general very difficult to read; copy editing for grammar is needed. A partial markup copy of what may be needed is available. As the specific GLKs are tied to specific expression profiles there is a need for added annotations to clearly define their attributes; the best way to do this is to present a table with gene ontology terms in a tabular form, and with the GO:12345 type annotations defining their biological, tissue, and molecular functions. I do see the transcripts characterized in Table S2; perhaps added columns here may help.

Journal manuscripts are often scanned by text-mining software that locates and extracts core data elements, like gene function. Adding standard ontology terms, such as the Gene Ontology (GO, geneontology.org) or others from the OBO foundry (obofoundry.org) can enhance the recognition of your contribution and description. This will also make human curation of literature easier and more accurate. GO: annotations help to define the biological, molecular, and cellular context of the gene.

The manuscript still appears in rough shape and requires additional revisions; please see markup (in a separate email) and seek improved copy editing."

---

## Round 0.4 · accepted · Accept

The manuscript is significantly improved.